# Cultural Adaptation and Validation of the Ambulatory Self-Confidence Questionnaire (ASCQ), Portuguese (European) Version

**DOI:** 10.3390/ijerph21081026

**Published:** 2024-08-04

**Authors:** Maria Teixeira, Mónica Luís, Magda Reis, Carlota Carvão, Anabela Correia Martins

**Affiliations:** 1Physiotherapy, Coimbra Health School, Polytechnic University of Coimbra, 3045-043 Coimbra, Portugal; a2020137437@estescoimbra.pt (M.T.); a2019134037@estescoimbra.pt (M.R.); a2020144593@estescoimbra.pt (C.C.); 2H&TRC—Health & Technology Research Center, Coimbra Health School, Polytechnic University of Coimbra, Rua 5 de Outubro, 3045-043 Coimbra, Portugal

**Keywords:** Portuguese validation, older adults, falls, ambulatory, self-confidence

## Abstract

In a world where physical activity and social participation are fundamental pillars of a full and healthy life, confidence in walking has emerged as a fundamental aspect to assess, especially for older adults. Therefore, the purpose of this study was to develop a Portuguese (European) version of the Ambulatory Self-Confidence Questionnaire (ASCQ) that was both linguistically and psychometrically adapted. To do so, a translation method was used, followed by an assessment of its validity and reliability. The Portuguese version was completed by 173 older adults. To assess reliability, Cronbach’s alpha and intraclass correlation coefficients (ICCs) were used. For sociodemographic and clinical characterization, as well as questionnaire scoring, descriptive statistical analysis was used. Pearson’s correlation (r), Student’s *t*-test, and one-way ANOVA were used to analyze criterion and construction validity. The Portuguese interactions with ASCQ were effectively translated and adjusted, revealing exceptional internal consistency and test–retest reliability, as reflected in Cronbach’s alpha and ICC values of 0.95. No floor effect was observed; however, a ceiling effect was identified (3.5%). The criterion and construct validity were verified as all the correlations established were statistically significant. The adaptation of the ASCQ to Portuguese culture is adequate, making it valid for use within the Portuguese population.

## 1. Introduction

With the development of technology and medicine, average life expectancy has increased and increasingly aging populations have emerged, especially in developed countries like Portugal, the second-oldest country in the European Union [1,2,3,4,5,6,7]. According to data published by PORDATA, around 24% of the Portuguese population is over 65, and there has been a significant increase since 2011 [4,6].

While the increase in longevity is appreciable, there is potential to improve the quality of the years of life gained. According to the literature, in Portugal, in 2015, at the age of 65, the number of healthy life years lost (DALYs) was 4.90 for women and 3.99 for men [4,6].

As stated by the World Health Organization (WHO), a fall, defined as an episode in which an individual inadvertently moves to the floor or another level lower than their initial position and which excludes collision with furniture, walls, or other structures, is undoubtedly one of the biggest problems among older adults [8,9,10]. Its prevalence among community-dwelling older adults is estimated to be approximately 30% each year [11,12,13,14,15]. Although most studies refer to people over the age of 65, people aged 50 and over often underestimate the risk of suffering a fall, yet it is the main reason for hospitalization in this age group [12].

Falls have multifactorial causes, including biological, environmental, behavioral, and socioeconomic factors [9,12,14,15,16,17].

Falls can have serious consequences for the individual and can have psychological impacts, such as fear of falling (FoF) and loss of confidence, as well as physical and social impacts, leading to reduced quality of life and high associated costs for health systems. These costs are estimated to be around EUR 25 billion per year in the European Union [11,12,14,16,18,19].

There is evidence that FoF, which is characterized by low perceived self-efficacy or confidence in avoiding falls during essential and non-hazardous activities of daily living, is very common among the elderly, affecting between 21% and 85% of older people who have already fallen and between 33% and 46% of those with no history of falls (HoF). It is also linearly related to frailty. Knowledge of this association is crucial in clinical practice, helping in the development of screening protocols, as well as primary or secondary preventions [20].

One of the main objectives in this population is to maintain mobility and ambulation, and one of the solutions that brings the most results is physical activity. At the same time, it also reduces the risk of injury and brings important benefits not only on a physical level but also on a psychosocial level [3].

As mentioned earlier, the progressive aging of the general population is one of the biggest challenges facing public health systems. As such, health professionals, such as physiotherapists, are crucial in preventing falls, although they are often underestimated [9,21]. It is essential to educate the community about the preventive role of these professionals and to intervene early to reduce the modifiable risk factors for falls, thus contributing to reducing years lived with disability.

As advocated by the American Geriatrics Society, the British Geriatrics Society, and the American Physical Therapy Association Geriatrics (APTA Geriatrics), regular screenings are essential to identify fall risk factors early on and to develop adapted exercise plans [16,22]. These screenings include some questionnaires to collect self-reported information and include some functional tests, namely the 10-meter walking speed, timed up-and-go, and 30 s sit-to-stand tests, in order to assess the health and mobility of older adults.

While most assessment instruments are based on quantitative measures, self-reported information is essential for understanding FoF and confidence when walking, which are important factors in preventing falls [23]. Therefore, there is a clear need for instruments that capture self-confidence during locomotion.

In 2007, Miller developed a questionnaire, the Ambulatory Self-Confidence Questionnaire (ASCQ), which consists of 22 questions that aim to capture a person’s self-confidence during ambulation. When it was developed, there was no instrument that assessed this aspect, which is why it has already been validated in several countries, allowing comparisons to be made between different populations, professionals from other countries to use it, and transnational studies to be conducted [24].

For the Ambulatory Self-Confidence Questionnaire to be used in Portugal, it needs to be adapted to our population and its psychometric characteristics assessed; thus, the aim of this study was to translate it, culturally and linguistically adapting and validating it for the Portuguese population.

According to the WHO, active aging involves optimizing opportunities for health, participation, and safety, improving quality of life and maintaining the functional capacity of older people [4,12,25,26,27]. Several studies have indicated that the frequency of older adults’ social activities during their retired life could eliminate feelings of loneliness and help them achieve a higher quality of life [26].

## 2. Materials and Methods

This study was conducted in two phases. The first involved the translation of the instruments into Portuguese (European), and the second included an assessment of psychometric properties.

### 2.1. Phase 1: Cultural and Linguistic Adaptation

The cross-cultural adaptation of the questionnaire was conducted in accordance with recommendations proposed by the Scientific Advisory Committee of the Medical Outcomes Trust [28]. Firstly, the ASCQ was translated into Portuguese (European) by two independent bilingual translators, considering both lexical and cultural equivalences. Both had Portuguese as their mother tongue. Subsequently, back-translation of each of the obtained translations was performed. Both translated versions were then discussed in a first academic consensus panel with extensive experience in the translation and back-translation of instruments, as well as knowledge of active and healthy aging interventions, whose members were instructed to check for meaning equivalences. Consensus was reached for all of the content, except for the expression “Ambulatory Self-Confidence”, which was translated to “Autoconfiança na Deambulação” and “Autoconfiança ao Andar”. After a brief debate, “Autoconfiança ao andar” was chosen because it is a clear and direct expression that uses common vocabulary that can be easily understood by most people. On the other hand, “Autoconfiança na Deambulação” is a more formal and technical expression, which may be less familiar to the general population. The result was a preliminary Portuguese (European) version of the ASCQ. Secondly, that version was administered to a pilot group of 8 people who were native and fluent in Portuguese. After completing the questionnaire, the subjects were invited to give their feedback on it to identify and correct potential difficulties. No doubts were raised regarding the clarity, comprehension, cultural relevance, or appropriateness of the words used in the translated version of the ASCQ.

### 2.2. Phase 2: Validation Study

This study was approved by the Ethics Committee of the Polytechnic Institute of Coimbra (Registration code 145_CEIPC_2023) and was conducted in accordance with the ethical principles of the Declaration of Helsinki. All of the participants provided their informed consent.

### 2.3. Sample

A sample of 173 community-dwelling adults (age 50 and older) was recruited to determine the reliability (internal consistency and test–retest reliability) and the validity of the Portuguese (European) ASCQ. This study included a convenience sample of adults aged 50 or over, living in the community, having the ability to complete the screening, and who took part in fall risk screening actions advertised in the usual places (municipalities and associations) in the center of Portugal. Participants were excluded from the study if they experienced difficulty in understanding the instructions for conducting the functional tests, raised doubts about their cognitive ability due to dementia and/or depression or, finally, despite their nationality, were not fluent in Portuguese.

### 2.4. Procedures

The data collection was conducted between February and March 2024 with the purpose of assessing the psychometric properties of the Portuguese adaptation of the ASCQ. After signing their informed consent, all eligible participants completed the FallSensing protocol’s questionnaires and the functional tests, to which was added the translated and adapted Portuguese (European) version of the ASCQ.

### 2.5. Reliability

According to the literature, the sample size was determined following the recommendations of having 4 to 10 participants for each item in the questionnaires [29].

Reliability was assessed in terms of internal consistency (Cronbach’s alpha) and test–retest reliability, by the intraclass correlation coefficients (ICCs).

To ascertain test–retest reproducibility, a sub-sample of 30 people signed a new informed consent form stating that they authorized and would be available to answer the ASCQ again by telephone 8 to 10 days after the face-to-face screening. In addition, it was also ensured that there were no health events during this period (accident, fall, or illness).

### 2.6. Content Validity

Content validity referred to whether the questionnaires were understood and whether all the important and relevant items had been included.

In addition to expert panel revision of the ASCQ items, the content validity was calculated using the floor and ceiling effects, whereby, if more than 15% of the participants provide minimum or maximum scores, the instrument is considered biased [30].

### 2.7. Criterion Validity

Criterion validity represents the accuracy with which a test measures the outcome it was designed to measure. A measuring instrument, such as a questionnaire, has criterion validity if its results converge with those of another instrument that has already been validated and is called the “gold standard” [31]. Therefore, criterion validity was assessed by comparing the mean total scores of the ASCQ with the FoF and the Profile of Activities and Participation related to Mobility (PAPM) and Exercise Self-Efficacy Scale.

### 2.8. Construct Validity

Construct validity refers to the accuracy with which a test measures the concept it was designed to measure. For this, hypotheses are usually generated, which are then tested to support the validity of the instrument [32]. Therefore, this validity was estimated through the use of several hypotheses that we established, namely that individuals that used a walking aid, had a greater HoF or frailty, and/or consumed four or more drugs, including benzodiazepines, would have lower scores on the ASCQ and that there is a correlation between individuals with lower self-confidence in walking and poorer functional tests.

### 2.9. Measures

To assess the validity of the ASCQ, measures from the FallSensing protocol were used. This includes several self-reported answers regarding HoF, FoF, self-perceived health, frailty, use of polypharmacy, including benzodiazepines; functional tests, such as 30 s sit-to-stand (30sSTS), 10-meter walking speed (10-MWS), and timed up-and-go test (TUG); and questionnaires, i.e., the Exercise Self-Efficacy Scale and PAPM. All of these measures were previously described in detail and were adopted in this study [16].

Ambulatory Self-Confidence Questionnaire (ASCQ)

The ASCQ consists of 22 items and aims to assess an individual’s self-confidence during ambulation in different environments and contexts in the community. The response options vary on a scale between 0 (no confidence) and 10 (total confidence). The average score is calculated (between 0 and 10) by adding up all the answers and then dividing by the number of questions. Higher scores indicate a higher level of confidence during ambulation [23,24].

### 2.10. Data Analysis

Data analysis was performed using descriptive statistics with IBM SPSS Statistics 29.0 for Windows.

For sociodemographic and clinical characterization, as well as questionnaire scores, descriptive statistical analysis was employed, including measures of central tendency (mean, M), dispersion (standard deviation, SD; maximum, Max; minimum, Min), and absolute (n) and relative (%) frequencies. Pearson’s correlation (r) was used for the analysis of correlations, and Student’s *t*-test and one-way ANOVA were used for the analysis of the differences between groups. Pearson’s correlation coefficient values were registered as follows: 0.9–1.0, very strong; 0.70–0.89, strong; 0.40–0.69, moderate; 0.10–0.39, weak; and 0.00–0.19, negligible [33]. Reliability was assessed using Cronbach’s alpha and intraclass correlation coefficient (ICC), and values above 0.70 and 0.75, respectively, are good and indicate good internal consistency and good test–retest reliability [34,35].

The interpretation of statistical tests was conducted based on a significance level of 0.05 (*p* ≤ 0.05), with a confidence interval of 95%.

## 3. Results

The Portuguese (European) version of the ASCQ was satisfactorily adjusted in terms of semantics, concept, language, and equivalence to the national culture.

### 3.1. Cultural and Linguistic Adaptation

The translation was conducted as planned. The content validity of the Portuguese (European) version of the ASCQ was achieved through collaborative revisions involving experts and older adults who took part in the screenings. During the screenings, all of the participants demonstrated a clear understanding of the questionnaire items. There were no difficulties in understanding any of the words or questions; thus, no changes were necessary. All of the participants provided answers to all items in the Portuguese (European) version of the ASCQ, and no missing items were found. The structure of the original version remained intact in the final adaptation of the ASCQ into Portuguese (European).

### 3.2. Psychometric Validation of the Portuguese (European) Version of the ASCQ

#### 3.2.1. Sample

A total of 173 community-dwelling adults aged 50 or over were included; 119 (68.8%) were female, with mean ages of 66.83 ± 8.7 (validity sample) and 65.73 ± 7.66 (reliability sample); 6.9% used a walking aid, 45.1% had fallen in the last 12 months, and more than half (52%) reported FoF. In addition, 35.8% of the individuals had a risk level of 3 (high risk of falling). The mean score of self-perceived health was 3.71 ± 1.0. Overall, 54.9% consumed four or more medications, including benzodiazepines (27.2%). Finally, 35.8% of the total were living alone and 15.0% experienced frailty (Table 1).

Regarding measure samples, the average scores were as follows: 12.82 on the scale of self-efficacy for exercise, 0.49 on the PAPM, 9.27 on the TUG, 13.54 on the 30sSTS, and, finally, 1.52 m/s on the walking speed, as assessed by 10-MWS (Table 1).

#### 3.2.2. Reliability

The total sample size consisted of 173 older adults, corresponding to 7.9 people per questionnaire item; therefore, this sample is within the size recommended in the literature. The Portuguese (European) version showed excellent internal consistency, as indicated by a Cronbach’s alpha coefficient of 0.95. The intergroup correlation coefficient (ICC) was 0.95 (*p* < 0.001).

#### 3.2.3. Content Validation

Content validity was first established through a review conducted by a panel of experts during the adaptation process and feedback from the population during the screening. The expert panel reached a consensus that the questionnaire presented all the relevant questions, and no additional questions were incorporated beyond those present in the original version. The target population indicated that the questionnaire was understandable. As can be seen in Table 2, all the items in the adaptation match those in the original version, and the average scores are also similar. None (0%) of the participants provided a minimum score, and six (3.5%) provided a maximum score to the ASCQ.

#### 3.2.4. Criterion and Construct Validity

The comparison of the ASCQ with the other variables as assessed through Pearson’s correlation, Student’s *t*-test, and one-way ANOVA is presented in Table 3.

Table 3 shows the confirmation of moderate correlations of the Portuguese version of the ASCQ with other questionnaires and that differences between the groups who reported being afraid of falling and not being afraid of falling were statistically significant. The Pearson correlation between the ASCQ and the PAPM was r = 0.691 (*p* < 0.001).

A lower ASCQ score was observed in individuals who used walking aids, had a HoF, had frailty, took four or more medications per day, took benzodiazepines, and had a higher risk level. All of these differences between groups were statistically significant. In addition, the correlation between the ASCQ and other measures was assessed using Pearson’s coefficient, with moderate correlations with the TUG (r = 0.609), 30sSTS (r = 0.423), 10-MWS (r = 0.531), and self-perceived health (0.453) being observed, as well as very strong correlations with the ASCQs answered by telephone (r = 0.909).

## 4. Discussion

The progressive aging of the population in general is one of the biggest challenges facing public health systems and has some negative aspects associated with it, including the increased risk of falling and the consequent decline in social participation and quality of life. Therefore, the development of effective fall prevention and management strategies is a priority worldwide, and health professionals, particularly physiotherapists, have or should have an active role in this process.

The ability of older adults to move around safely and independently is one of the most important skills for maintaining a good quality of life and requires a combination of many different functions and skills, including physical, psychosocial, and environmental conditions [22]. Therefore, measuring ambulatory confidence can be important. To do this, there need to be instruments that can help assess this parameter. Until 2007, there were few or no instruments that gave us self-reported information, which is why Dr. Miller developed the Ambulatory Self-Confidence Questionnaire (ASCQ) [23]. However, for this to be viable in a clinical and research context in Portugal, it needs to be validated and adapted linguistically and culturally for our population.

On the linguistic side, it was crucial to maintain the original context, meaning, instructions, and presentation of the questionnaire to ensure adequate equivalence. The Portuguese (European) version of the ASCQ received excellent agreement among experts regarding the relevance of the questions. It underwent revisions by independent bilingual translators and was pilot-tested, revealing no inconsistencies and confirming its excellent content validity. As expected, the reliability of the Portuguese (European) version of the ASCQ was confirmed by the excellent Cronbach’s alpha and ICC values. As confirmed by the ceiling and floor effects, the ASCQ is not biased [30].

There were correlations between ASCQ, PAPM, and self-efficacy for exercise, as well as differences between the groups who reported being afraid of falling and not being afraid of falling. This difference is justified by the fact that fear of falling leads to limitations in activities, which, in turn, lead to restrictions in social participation and the perception of decreased health, consequently resulting in lower self-confidence [16]. These results proved the questionnaire’s criterion validity, which demonstrates that the ASCQ is an accurate instrument and measures the outcome it was designed to measure, namely confidence in walking [31].

Regarding construct validity, the previously established hypotheses were confirmed, i.e., that there were differences between the groups of those who had and those who did not have a HoF. The rationale for these hypotheses is based on the evidence that HoF is a risk factor for falling, and that it is the strongest predictor of future falls [16].

Another hypothesis that was confirmed was the association between the ASCQ and taking four or more medications, including benzodiazepines. There is evidence that approximately 20.3% of adults aged 55 or over who take four or more medications, or older people who take more than three or four medications, especially antidepressants and benzodiazepines, have an increased risk of recurrent falls [36,37]. Benzodiazepines can induce symptoms such as dizziness, confusion, memory loss, vision impairment, and loss of coordination, resulting in an increased risk of falling, and decreased ambulatory self-confidence [38].

In addition to these, there was also an association between ASCQ score and frailty, justified by the fact that frailty and fear of falling are linearly related and, as such, individuals who are frailer have less self-confidence in walking, which is reflected in the average score on the questionnaire [20]. It was also evident that individuals who used walking aids [39,40] and who had a higher level of risk [41,42] had lower scores on the questionnaire.

Finally, regarding the functional tests, there was a moderate correlation between the ASCQ and the TUG, the 30sSTS, the 10-MWS, and self-perceived health, and a very strong correlation with the ASCQ completed at home, 8 to 10 after the face-to-face screening.

The strengths of this study are linked to the size of the sample, which is within the parameters recommended by the literature for this type of study, and the commitment and collaboration of the sub-sample who were invited to answer the questionnaire a few days after the face-to-face screening, with 100% adherence. On the other hand, with regard to limitations, one of them is related to recruitment bias, i.e., the fact that the sub-sample used to assess test–retest reproducibility was invited by the researcher for convenience, and another, also related to the sub-sample, is the fact that the ASCQ baseline data for the reliability analysis were collected in the presence of the researcher, unlike the follow-up data. Although the researcher did not interfere in filling in the ASCQ, there was no way of controlling the influence of the participant’s friends or family on the follow-up.

## 5. Conclusions

The results of this study confirm the reliability and support the validity of the Portuguese (European) version of the ASCQ for community-dwelling older adults. The Portuguese (European) version of the ASCQ has semantic, conceptual, idiomatic, and content equivalence when compared to the original version. The criterion validity and construct validity were proved. One of the most important skills for maintaining high levels of functioning and social participation in an aging population is promoting the ability and confidence to walk safely and independently. This questionnaire, designed to assess self-confidence in walking, is a useful and easily applicable tool in a clinical context, in a community-based context, and for scientific research in several fields, namely in active and healthy aging.

## Figures and Tables

**Table 1 ijerph-21-01026-t001:** Sociodemographic and functional data of the validity and the reliability samples.

Characteristics	Validity Sample (N = 173)	Reliability Sample (N = 30)
Age, mean ± SD	66.83 ± 8.7	65.73 ± 7.7
Woman, N (%)	119 (68.80)	19 (63.3)
Walking aid, N (%)		
None	161 (93.1)	30 (100)
Cane	6 (3.5)	-
Crutches	5 (2.9)	-
Sticks	1 (0.6)	-
≥4 medications, N (Yes%)	95 (54.9)	16 (53.3)
Benzodiazepines, N (Yes%)	47 (27.2)	12 (40)
Highest education completed, N (%)		
1stcycle	66 (38.2)	11 (36.7)
2nd cycle	24 (13.9)	-
3rd cycle	32 (18.5)	7 (23.3)
Secondary	25 (14.5)	8 (26.7)
Degree	4 (2.3)	2 (6.7)
Master	5 (2.9)	1 (3.3)
Doctorate	5 (2.9)	-
Living alone, N (%)	62 (35.8)	9 (30)
Frailty, N (%)	26 (15.0)	2 (6.7)
Risk level, N (%)		
1 Low	54 (31.2)	10 (33.3)
2 Moderate	57 (32.9)	13 (43.3)
3 High	62 (35.8)	7 (23.3)
Self-perceived health, mean ± SD	3.71 ± 1.0	4.47 ± 0.9
History of falls (the past 12 months), N (%)	78 (45.1)	14 (46.7)
Fear of falling, N (Yes%)	90 (52.0)	14 (46.7)
Self-efficacy for exercise, mean ± SD	12.82 ± 4.9	13.40 ± 3.9
PAPM, mean ± SD	0.49 ± 0.8	0.13 ± 0.3
TUG, mean ± SD	9.27 ± 3.3	7.40 ± 2.2
30sSTS, mean ± SD	13.54 ± 4.3	15.30 ± 4.6
10-MWS, mean ± SD	1.52 ± 0.4	1.66 ± 0.4

Abbreviations: PAPM: Profile of Activities and Participation related to Mobility; TUG: timed up-and-go; 30sSTS: 30 s sit-to-stand; 10-MWS: 10-meter walking speed; SD: standard deviation.

**Table 2 ijerph-21-01026-t002:** Descriptive data for the ASCQ items, validity sample (Portuguese version and original), and reliability sample.

Item		Validity Sample (N = 173)	Realibility Sample (N = 30)
		Mean ± SD	Mean ± SD
1	Subir para um passeio?	8.25 ± 2.6	9.13 ± 1.4
	Step up onto a curb?	8.79 ± 2.4	
2	Descer de um passeio?	8.05 ± 2.6	8.93 ± 1.5
	Step down off a curb?	8.57 ± 2.4	
3	Subir uma rampa (inclinação ligeira)?	7.65 ± 2.8	8.57 ± 1.8
	Walk up a ramp (mild incline)?	9.40 ± 1.6	
4	Descer uma rampa (inclinação ligeira)?	7.53 ± 2.7	8.30 ± 1.9
	Walk down a ramp (mild incline)?	9.26 ± 1,7	
5	Subir um lanço de escadas (4 degraus ou mais) com um corrimão?	8.02 ± 2.6	8.80 ± 1.4
	Walk up a flight of stairs (4 steps or more) with a handrail?	8.99 ± 2.3	
6	Descer um lanço de escadas (4 degraus ou mais) com um corrimão?	7.88 ± 2.6	8.70 ± 1.5
	Walk down a flight of stairs (4 steps or more) with a handrail?	8.79 ± 2.3	
7	Atravessar uma rua, numa passadeira, com semáforo de peões (cronometrado)?	8.79 ± 2.3	9.30 ± 1.1
	Cross a street with a timed crosswalk (walk signal)?	9.25 ± 1.7	
8	Atravessar uma rua, numa passadeira, sem semáforo de peões (cronometrado)?	8.10 ± 2.6	8.73 ± 1.2
	Cross a street without a timed crosswalk (walk signal)?	8.57 ± 2.5	
9	Andar num passeio desnivelado?	6.82 ± 2.9	7.47 ± 2.0
	Walk on an uneven sidewalk?	7.96 ± 8.0	
10	Andar na relva?	8.61 ± 2.5	9.27 ± 1.3
	Walk on grass?	8.45 ± 2.5	
11	Andar num pavimento escorregadio?	4.16 ± 3.1	5.70 ± 2.2
	Walk on slippery ground?	6.12 ± 3.2	
12	Andar no escuro ou à noite quando é difícil ver os seus pés?	5.34 ± 3.7	6.70 ± 2.4
	Walk in the dark or at night when it is difficult to see your feet?	7.51 ± 2.9	
13	Atravessar um lugar com muita gente?	7.37 ± 3.1	8.50 ± 1.9
	Walk through a crowded place?	8.34 ± 2.3	
14	Andar e falar ao mesmo tempo com um acompanhante?	8.18 ± 2.6	8.63 ± 2.2
	Walk and talk to a companion at the same time?	8.60 ± 2.3	
15	Carregar pequenos objetos enquanto caminha?	7.65 ± 3.1	8.53 ± 2.3
	Carry small items while walking?	8.60 ± 2.7	
16	Parar de andar de forma súbita para evitar um veículo que vem na sua direção?	7.80 ± 3.1	8.57 ± 1.9
	Stop walking suddenly to avoid an oncoming vehicle?	8.67 ± 2.2	
17	Usar uma escada rolante?	7.72 ± 3.3	8.50 ± 1.5
	Use an escalator?	8.51 ± 2.8	
18	Usar um tapete rolante em movimento?	7.82 ± 3.2	8.77 ± 1.3
	Use a moving sidewalk?	8.43 ± 2.8	
19	Movimentar-se dentro de um autocarro em movimento?	6.20 ± 3.6	6.40 ± 3.0
	Walk on a moving bus?	6.86 ± 3.2	
20	Ir de uma divisão a outra na sua casa?	9.57 ± 1.3	9.83 ± 0.5
	Walk from one room to another in your home?	9.61 ± 1.2	
21	Andar uma distância curta sem parar?	9.46 ± 1.4	9.73 ± 0.7
	Walk a short distance without stopping?	9.35 ± 1.9	
22	Andar uma distância longa sem parar?	8.65 ± 2.3	8.70 ± 2.0
	Walk a long distance without stopping?	8.74 ± 2.7	
Total ASCQ score		7.70 ± 2.0	8.44 ± 1.1
		8.52 ± 1.7	

**Table 3 ijerph-21-01026-t003:** Associations and differences between groups to assess the criterion validity and the construct validity.

	ASCQ (N = 173)			
	Mean ± SD	r	Z	*p*
Walking aid				
Yes (n = 14)	5.50 ± 2.5			≤0.001 ^1^
No (n = 159)	7.90 ± 1.8			
History of falls (the past 12 M)				
Yes (n = 78)	7.17 ± 2.1			≤0.001 ^1^
No (n = 95)	8.14 ± 1.8			
Fear of falling				
Yes (n = 90)	6.89 ± 2.1			≤0.001 ^1^
No (n = 83)	8.59 ± 1.4			
Frailty (weight loss)				
Yes (n = 26)	6.52 ± 2.7			≤0.001 ^1^
No (n = 147)	7.92 ± 1.8			
≥4 Medication				
Yes (n = 95)	7.14 ± 2.1			≤0.001 ^1^
No (n = 77)	8.38 ± 1.7			
Benzodiazepines				
Yes (n = 47)	7.70 ± 1.7			≤0.005 ^1^
No (n = 81)	8.32 ± 1.4			
Risk level				
1 Low (n = 54)	8.85 ± 1.38		25.04	≤0.001 ^2^
2 Moderate (n = 57)	7.87 ± 1.76			
3 High (n = 62)	6.55 ± 2.04			
TUG		0.609		≤0.001 ^3^
30sSTS		0.423		≤0.001 ^3^
10-MWS		0.531		≤0.001 ^3^
PAPM		0.691		≤0.001 ^3^
Self-efficacy for exercise		0.455		≤0.001 ^3^
Self-perceived health		0.453		≤0.001 ^3^
ASCQ (answered by telephone) (N = 30)		0.909		≤0.001 ^3^

Abbreviations: TUG: timed up-and-go; 30sSTS: 30 s sit-to-stand; 10-MWS: 10-meter walking speed; PAPM: Profile of Activities and Participation related to Mobility; ASCQ: Ambulatory Self-Confidence Questionnaire; SD: standard deviation. Significance value of ^1^ Student’s *t*-test, ^2^ one-way ANOVA, and ^3^ Pearson’s correlation.

## Data Availability

The data presented in this study are available upon request from the corresponding author due to privacy and ethical reasons.

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
