# Peer review of "Cultural Adaptation and Validation of the Ambulatory Self-Confidence Questionnaire (ASCQ), Portuguese (European) Version"

_ijerph, 2024, doi:10.3390/ijerph21081026_

Round 1

Reviewer 1 Report

Comments and Suggestions for Authors

This manuscript reported reliability and validity in the instrument of ASCQ of the Portuguese version. There are several suggestions to the authors, to add more content precisely for reading.

 1. This article needs to explain the significance and purpose of developing and validating the cultural adaptation of this scale.

2. In the method of 2.1. stated that in Phase 1 by using translation and back translation to check the Cultural and Linguistic Adaptation of this scale. However, it needs to describe the different parts between this Portuguese version and the original language version.

The suggestion to the author: the translation of the ASCQ scale to the Portuguese version needs to provide instructions on how to make the instrument's simplicity and clear formulation of the sentences, as well as the conceptual, cultural, semantic, and idiomatic agreement.

 3. This paper stated that "A sample of 173 community-dwelling adults (age 50 and older) was recruited to determine the reliability (internal consistency and test-retest reliability) and the validity of the Portuguese ASCQ."

In the original study in developing ASCQ (Asano, Miller, Eng, 2007) , the criteria to participate in the study participants had to be:  >= 65 years of age; have minimal cognitive impairment, and be capable of walking a minimal distance (10 m) with or without a walking aid. Therefore, the criteria of participants in this study may need to provide more measurement data.

4. For this study, what rule is used to estimate the sample size? For the purpose of the study, the minimum number of participants was estimated at least 220, complying with the target of 10 participants for each scale item.

5. To assess the construct validity, the Timed Up and Go Test (TUG), Exercise Self-Efficacy Scale, PAPM17, and ASCQ scales were administered to assess hypothesized relationships with the ASCQ. However, the instrument for assessing construct validity should include more information on criterion instruments in section 2.9.

6. Measuring ambulatory confidence can be crucial. However, there is little to no information available on criterion instruments in clinical and research contexts in Portugal. It is essential to validate this information of this population.

Comments on the Quality of English Language

Minor editing of English language required.

Reviewer 2 Report

Comments and Suggestions for Authors

Line 122: the use of the age of 50 must be better explained. Major ageing neuromuscular changes occurs after the age of 65. Using a Middle Aged category might induce a bias.  Is there any results focusing of only older adults. 

Line 138. A measure of sample size could be more precise. 

The presentation of results could be better adapted. The statistics need to be better explained as not well represented in the table. 

A suggestion would be to divide in age categories, especially for some categories: frailty for example, which occur normally way after 65.

Comments on the Quality of English Language

Overall, sentences need a major English review. Many grammatical errors.
